# Are Supplements Safe? Effects of Gallic and Ferulic Acids on In Vitro Cell Models

**DOI:** 10.3390/nu12061591

**Published:** 2020-05-29

**Authors:** Francesca Truzzi, Maria Chiara Valerii, Camilla Tibaldi, Yanxin Zhang, Veronika Abduazizova, Enzo Spisni, Giovanni Dinelli

**Affiliations:** 1Department of Agricultural and Food Sciences, Alma Mater Studiorum–University of Bologna, 40127 Bologna, Italy; francesca.truzzi3@unibo.it (F.T.); camilla.tibaldi2@unibo.it (C.T.); yanxin.zhang2@unibo.it (Y.Z.); nika_abduazizova@mail.ru (V.A.); 2Department of Biological, Geological, and Environmental Sciences, Alma Mater Studiorum–University of Bologna, 40126 Bologna, Italy; chiaravalerii@hotmail.it (M.C.V.); enzo.spisni@unibo.it (E.S.)

**Keywords:** supplements, gallic acid, ferulic acid, intestinal wall models

## Abstract

Polyphenols display health-promoting properties linked to their biological activities. They are initially absorbed in the small intestine, then they are largely metabolized in the colon, whereupon they are able to exert systemic effects. The health-promoting properties of polyphenols have led to the development of food supplements, which are also largely consumed by healthy people, even if data on their safety are still yet lacking. In the present paper, the content of gallic acid and ferulic acid was analyzed in two supplements, and shown to be higher than the relative contents found in fruit and flour. To evaluate the effects of these phenolic compounds on epithelial intestinal tissue, gallic and ferulic acids were added to a new in vitro model of the intestinal wall at different concentrations. The effects on viability, proliferation and migration of these compounds were respectively tested on three different cell lines (Caco2, L929 and U937), as well as on a tridimensional intestinal model, composed of a mucosal layer and a submucosa with fibroblasts and monocytes. Results indicated that gallic and ferulic acids can exert toxic effects on in vitro cell models at high concentrations, suggesting that an excessive and uncontrolled consumption of polyphenols may induce negative effects on the intestinal wall.

## 1. Introduction

Polyphenols are natural compounds, chemically characterized by phenolic structural units. They are known to have health properties linked to their biological activities [1]. These compounds are mainly produced by plants as secondary metabolites, and act as antioxidants by reacting with reactive oxygen species (ROS). Moreover, polyphenols may be involved in gene regulation on a cellular level [2]. Polyphenols are divided into two principle subgroups: flavonoids (flavones, isoflavones, flavonols, flavanones, flavan-3-ols, anthocyanins and condensed tannins), with a primary structure consisting of two benzene rings connected by a pyrone C-ring, and non-flavonoids (benzoic acids, stilbenes, lignans, gallotannins, hydrolysable tannins and ellagitannins), with more heterogeneous and complex structures [3]. Phenolic compounds are not readily available, since only a small percentage is digested and adsorbed in the small intestine. They are predominantly metabolized by gut microbiota in the colon. Resulting polyphenol metabolites are then adsorbed, after which they are known to exert a systemic effect [4]. Polyphenols have been shown to perform free radical scavenging activities, related to their specific structural features, such as the number and position of hydroxyl and catechol units [5]. Due to these structural properties, polyphenols are known to reduce the prevalence of several oxidative stress markers, and thereby result in their beneficial roles in some pathological conditions, such as cardiovascular disease and diabetes [6,7]. An effect on the large bowel has also clearly been demonstrated, since polyphenols are able to promote healthy microbiota composition [8]. Microbiota modulation, in particular, has been linked to the ability of phenolic compounds to counteract colitis, induced by Dextran sodium sulphate, in mice [9]. A direct effect on intestinal cells has also been demonstrated. Caffeic acid and chlorogenic acid were reported to suppress TNF-α and IL-8 secretion in Caco2 cells [10]. Among the activities related to polyphenols, pro-oxidant activities [11] have also been reported. For instance, the spontaneous oxidation of gallic acid has been shown to generate several highly reactive species, including O_2_^−^, H_2_O_2_, quinones and semiquinones [12], and the generation of these species was dependent on both the polyphenol concentration and the environment [13,14]. However, phenolic compounds have been primarily considered for their health-promoting properties. As a result, this has led to the development of an increasing number of food supplements, which are also largely ingested by healthy people to prevent non-communicable disease development and, more generally, to promote well-being. To date, however, no regulatory legislation exists to regulate polyphenol contents in food supplements, and data on the safety of these compounds are still lacking.

Although in vitro models are not optimal for evaluating the systemic effects of phenolic compounds, since they are largely metabolized by intestinal microflora and then adsorbed, a direct effect on the intestinal epithelium before their metabolization cannot be excluded. In order to evaluate the interaction between phenolic compounds and epithelial intestinal tissue, we developed a tridimensional in vitro model of colon tissue, and tested the effects of two of the non-flavonoid compounds, namely gallic acid and ferulic acid, on mucosal integrity, as well as their interaction with submucosal cellular components. The effects of these compounds were also tested on three different cell lines (Caco2, L929 and U937) to assess their effects on cell viability, toxicity, migration and apoptosis. Results indicated that gallic and ferulic acids can exert differing effects on cell models, according to their concentration and the cell type analyzed. However, they showed toxic effects in in vitro cell models at high concentrations, suggesting that an excessive and uncontrolled consumption of polyphenols could induce negative effects on intestinal mucosa.

## 2. Materials and Methods

### 2.1. Chemicals

3-(4,5-dimetiltiazol-2-il)-2,5-difeniltetrazolio (MTT) assay was from Life Technologies (Carlsbad, CA, USA). Gallic acid, with purity of ≤ 100%, was from Sigma-Aldrich (Milan, Italy) and ferulic acid, with purity of ≥ 99%, was from Extrasynthese (Genay, France). Reagents for cell cultures, such as Hank’s Balanced Salt Solution (HBSS), Dulbecco’s Modified Eagle Medium (DMEM), Fetal Bovine Serum (FBS), L-Glutamine, Penicillin-Streptomycin, RPMI-1640 Medium and rat tail collagen type I were purchased from GIBCO (Waltham, MA, USA). Methanol and water (HPLC-grade, Lichrosolv^®^) were purchased from Merck (Darmstadt, Germany). All other chemicals and solvents were of analytical grade. The food supplements were based on the presence of a single raw material in addition to some excipients. The raw material of the first one was a dry extract obtained from an apple cultivar (Annurca), the second one was based on dry blueberry extract.

### 2.2. HPLC-MS/MS Analysis

Sample preparation: Standard stock solutions of gallic acid and ferulic acid, both at a concentration of 1 mg/mL, were prepared in water/methanol mixture (50/50, *v*/*v*). Working standard solutions were made by diluting stock solutions in water. Stock working solutions of the standards were stored at −20 °C. The extraction method used was as follows: 2 mL of methanol/acetone/water mixture (60/10/30, *v*/*v*/*v*) was added to 0.5 g of sample (dietary supplement). The mixture was stirred for 30 min and then centrifuged at 10,000 rpm for 10 min at 15 °C. The supernatant was filtered through a 0.22 μm nylon filter. Before injection into the HPLC-MS (High Performance Liquid Chromatography), a blueberry-based dietary supplement and dietary supplement A were diluted (1:5 and 1:1, respectively) in water.

Sample analysis: Samples were injected into a Waters e2695 Alliance HPLC System, coupled with a Waters ACQUITY QDa Mass Detector, as described by [15,16]. A Supelco_Analytical (Sigma-Aldrich St Louis, MO, USA) C18 15 cm × 4.6 mm, 5μm was selected. The mobile phase consisted of (A) water + 0.1% acetic acid and (B) methanol. A gradient experiment was performed from 5% to 10% B in 4.17 min, from 10% to 20% B in 4.16 min and from 20% to 95% B in 14.67 min, respectively. Solvent B returned to 5% in 4 min and was maintained for 8 min to re-equilibrate the column. The flow rate, the injection volume and column temperature were 0.86 mL/min, 23.5μL and 40 °C, respectively. The ionization source was used in the Electrospray Ionization ESI negative mode, using single ion recording (SIR). Optimal cone voltage was set at 15 V and the capillary voltage at 0.8 kV. The monitored (M–H)-ions were *m*/*z* 168.88 for gallic acid, and *m*/*z* 192.94 for ferulic acid. Data acquisition, data handling and instrument control were performed using the Empower 3 software.

### 2.3. Cell Model Systems

L929 mouse fibroblasts (ATCC-CCL1) were cultured with DMEM, to which 10% fetal bovine serum, 1 mM L-glutamine and 1% penicillin-streptomycin were added. The Caco2 human epithelial cell line (ATCC HTB-37), obtained from Colorectal adenocarcinoma, was cultured with DMEM, supplemented with 10%, fetal bovine serum and 1% penicillin-streptomycin. U937, a pro-monocytic, human myeloid leukemia cell line (ATCC CRL-1593.2), was cultured in RPMI-1640 Medium, supplemented with 10% fetal bovine serum and 1% penicillin-streptomycin. Stock solution of gallic and ferulic acids were prepared at the concentration of 1 mg/L in ethanol 70% *v*/*v*. Cells were treated, 24 h after seeding, with the following concentrations of gallic and ferulic acids: 2.5, 5, 10, 20 and 40 mg gallic acid equivalents (GAE)/L, respectively. The latter were diluted in DMEM for 24 h, after which their effects were evaluated in terms of cytotoxicity. Ethanol (70% *v*/*v*) was used as the control, suspended in DMEM alone.

### 2.4. MTT Viability Assay

Cell lines were treated either with five concentrations of gallic and ferulic acids, respectively, for 24 h, or in 70% ethanol as the control (suspended in DMEM alone). Adherent cells, as well as the L929 and Caco2 cells, were plated onto 96-well tissue culture plates (10^5^ cells/well) in complete medium. After treatments, proliferative cells were detected using the MTT assay, according to the ISO 10993-5 International Standard procedure (ISO 10993-5, 2009). The MTT substrate was prepared in DMEM, then added to cells in culture to attain a final concentration of 1 mg/mL, and then incubated for 2 h in the culture incubator at 37 °C with 5% CO_2_. After incubation, the medium was removed by aspiration. Isopropanol (100 μL) was added to each well and formazan dye formation was evaluated by a multi-well scanning spectrophotometer at 540 nm. Suspension cells, of the U937 cell line, were plated at a concentration of 10^5^ cells/well onto 96-well culture plates in complete medium. After treatments, the cells were incubated with 0.5% MTT solution in PBS for 4 h at 37 °C, and then dissolved with 100 μL isopropanol in 0.04 N HCl. The plate was read at 540 nm and results were expressed as percentage of viable cells with respect to untreated controls (70% ethanol). The percentage of cell proliferation was calculated using the following formula: (absorbance value of treated sample/absorbance value of control) × 100 = % of cell viability.

### 2.5. Cell Viability

Cell viability was measured using the blue trypan assay. The L929, Caco2 and U937 cells were each plated onto a 24-well tissue culture plate (5 × 10^6^ cells/well) in complete medium. After 24 h, treatments diluted in DMEM were added to cells for another 24 h. To detect viability, cells were then carefully resuspended in a 0.4% Trypan Blue (Gibco) solution and vital cells were counted using Countess^®^II FL (ThermoFisher Scientific, Waltham, MA, USA). The results were expressed as a viability percentage of the control.

### 2.6. Caspase3/7 Detection

The L929, Caco2 and U937 cells were each plated onto a 24-well plate (10^4^ cells/well). Both treated and untreated cells were resuspended in CellEvent™ Caspase-3/7 Green Detection Reagent (ThermoFisher Scientific) for 30 min in PBS with 20% of FBS, according to a data sheet protocol. Cells were resuspended and analyzed using the Countess II FL Automated Cell Counter (ThermoFisher Scientific). Each sample was analyzed in duplicate.

### 2.7. Wound-Healing Assay

A total of 2 × 10^5^ L929 fibroblasts were plated onto 24-well tissue culture plate. At 24 h after seeding, cells were washed three times in HBSS and a line for each well was drawn along the cell monolayer with a sterile plastic tip. Plates were washed twice with HBSS to remove all detached cells and incubated in DMEM with 5 mg gallic acid equivalents (GAE)/L gallic or ferulic acids or 70% ethanol as the control. Cells were monitored at 4 and 24 h from stimulation and pictures of cells were taken. The results of each experiment were expressed as the mean of migrated cells from three different areas. The final results were expressed as the mean of three different experiments.

### 2.8. Monocyte Migration

For the monocyte migration experiment, 50 × 10^4^ U937 in each well was placed on the top of a 0.4 μm filter of tissue culture inserts (Transwell, Costar, Cambridge, MA, USA) in 12-well plates. On the bottom of each well, 2 × 10^5^ Caco2 cells were either plated or not plated. After a period of 24 h, Caco2 cell medium was replaced with DMEM together with 20 mg/L of either gallic or ferulic acid. After 4 h, filter supports were removed, stained with thiazine for just 10 s and cell migration counted manually using a microscope (6 fields for each condition). The experiment was repeated 3 times.

### 2.9. Intestinal Equivalents

For intestinal equivalents, 0.5 mL of a cell free collagen solution (1.35 mg/mL rat tail collagen type I in DMEM with 10% FCS and 1% Pen/Strep) was added to tissue culture inserts (Transwell, Costar, Cambridge, MA, USA) in 12-well plates. This pre-coated layer was overlaid with 1 ml of L929 fibroblasts (10^5^/mL) together with monocytes (3*10^4^/mL) mixed with collagen type I. After 2 h of incubation at 37 °C, 2 × 10^5^ of Caco2 cells were seeded onto dermal reconstructs, and incubated at 37 °C with Caco2 medium, added both on the upper and the lower part of the filter support. After 5 days, intestinal equivalents were either treated or not treated with 5 or 20 mg/L of either gallic or ferulic acid, or 70% ethanol for the control, respectively. The intestinal equivalents were then fixed with formalin for 2 h at room temperature, dehydrated and embedded in paraffin.

### 2.10. Immunohistochemical Analysis

Paraffin-embedded intestinal equivalents were rehydrated. Sections (4 μm thick) were stained with hematoxylin and eosin (H&E) and thiazine staining (Bio-Optica^®^, Milan, Italy). Occludin tissue expression was detected using a mouse anti-occludin primary antibody (Novus). Immunohistochemistry was performed using Fast Red chromogen. Stainings were performed according to the UltraVision LP Detection System AP Polymer & Fast Red Chromogen assay (Thermo Fisher Scientific, Waltham, MA, USA). Negative controls were obtained by omitting the primary antibody. The expression intensity was quantitatively determined using ImageJ software (Wayne Rasband). Signal detection was performed on at least 3 different sections and the results were reported as a mean.

### 2.11. Statistical Analysis

All the statistical analyses were performed using the Statistica 7.1 software (2005, StatSoft, Tulsa, OK, USA). The Bartlett’s test, which verifies the basic assumption of ANOVA as the homogeneity of variances, was statistically significant (*p* > 0.05). The Duncan test, which is adequate in the case of non-homogeneity of varieties, was employed as a post hoc test. Linear discriminant analysis (LDA) was performed on a standardized matrix of the MTT results, viability and cell type tests, and included a total of 5 variables, namely: MTT activities and viability results on L929, Caco2 and U937 treated with firstly gallic acid, 2.5 (G2.5), 5 (G5), 10 (G10), 20 (G20) and 40 (G40) mg/L, and secondly, with ferulic acid 2.5 (F2.5), 5 (F5), 10 (F10), 20 (F20) and 40 (F40) mg/L; migration analysis on L929 treated firstly with gallic acid, 2.5 (G2.5), 5 (G5), 10 (G10), 20 (G20) and 40 (G40) mg/L, and secondly with ferulic acid, 2.5 (F2.5), 5 (F5), 10 (F10), 20 (F20) and 40 (F40) mg/L. Linear discriminant analysis is a multivariate technique that allows the scoring of cases as a function of the first two roots (canonical discriminant functions). This technique was used to visualize similarities and differences among genotypes [17]. It is also a statistical method used to find a linear combination of features (discriminant functions) that characterizes or separates two or more classes of objects or events. Linear discriminant analysis (LDA) was performed using Statistica 7.1 software (2005, StatSoft, Tulsa, OK, USA). The supervised learning technique was applied to the standardized data matrix of responses (vitality, migration and respiration MTT test) of the cell lines (Caco2, L929, U937) as a function of different concentrations of gallic and ferulic acids. The cases (gallic and ferulic acid at the doses of 0, 2.5, 5, 10, 20 and 40 mg/L) were scored according to the first two roots (canonical discriminant functions).

## 3. Results

### 3.1. Gallic and Ferulic Acid Content in Commercially Available Dietary Supplements

The two polyphenols, gallic and ferulic acids, which are highly expressed in fruits and cereals, respectively, were quantified in two commercially available dietary supplements, referred to as “Blueberry supplement” and “Supplement A”, the last one obtained from apple polyphenols extract. The objective of this study has been to understand the quantity of polyphenols that can be ingested via common dietary supplements. In both dietary supplements analyzed, gallic acid was present in a lower concentration compared to ferulic acid (Table 1). In particular, Blueberry supplement showed a concentration of gallic and ferulic acids of 0.3252 mg/g and 1.3695 mg/g, respectively. Supplement A displayed a concentration of gallic and ferulic acids of 0.6354 mg/g and 1.8494 mg/g, respectively (Table 1, Appendix A).

### 3.2. Effects of Gallic and Ferulic Acids on Mouse Fibroblast Proliferation and Migration

In order to understand the effects of gallic and ferulic acids on the proliferation and wound-healing ability of cellular components within the intestinal dermis, their effects on the L929 fibroblast’s cell line have been studied. The cells were treated with increasing doses of gallic acid and ferulic acid (2.5, 5, 10, 20 and 40 mg/L), respectively, diluted in DMEM. The two phenolic acids showed differing effects, depending on the dose. No significant effect, attributable to the two phenolic acids at any dosage, was detected on cellular proliferation compared to the control (Figure 1a,b). Regarding the viability tests, a significant toxic effect was elicited by both gallic and ferulic acids at 40 mg/L (Figure 1c,d). Caspases 3/7 was not activated, by either gallic acid or ferulic acid stimulation, at 20 mg/L, compared to the untreated control (Figure 1e). The wound-healing assay showed that different concentrations of the two phenolic acids could either favor or not favor the shelter of intestinal tissue lesions. Cell migration was monitored over a period of 48 h, starting 4 h after addition of the stimulus, and terminating 48 h later. The most interesting results were observed 24 h after the treatments (Figure 2c). Ferulic acid showed a significant stimulating effect at 5, 10 and 20 mg/L, compared to the control, whereas a dose of 40 mg/L inhibited wound-healing (Figure 2a). Regarding gallic acid, 2.5 and 5 mg/L induced fibroblast migration, whereas this effect was inhibited at 40 mg/L (Figure 2b).

### 3.3. Effects of Gallic and Ferulic Acid on Human Intestinal Cell Viability

In order to understand the effects of gallic and ferulic acids on the proliferation and viability of cellular components which belong to the intestinal wall mucosa, their effects on the Caco2 colonocyte cell line have been studied. Gallic acid significantly reduced cell proliferation at 40 mg/L (Figure 3a), as well as cell viability at both 20 and 40 mg/L. In contrast, ferulic acid reduced cell proliferation (Figure 3b) and cell viability (Figure 3d) at 10 and 20 mg/L, respectively, but not at 40 mg/L. The quantification of the effector Caspases 3/7, following the treatment of Caco2 with gallic and ferulic acids, at 20 mg/L (Figure 3e) and 40mg/L (Figure 3f) respectively, for 24 h, showed a significant activation, compared to the control (Figure 4d). An increased effect was elicited by gallic acid.

### 3.4. Effects of Gallic and Ferulic Acids on Human Monocyte Cell Viability

To understand the effects of gallic and ferulic acids on the viability of the immunity cellular components of the intestinal dermis compartment, their effects on the U937 monocyte cell line have been studied. Interestingly, gallic acid significantly up-regulated monocyte proliferation at 5, 20 and 40 mg/L, compared to the untreated control (Figure 4a). Cell viability remained unchanged, and was 100% upon each treatment analyzed (Figure 4c). Ferulic acid had no significant effect on U937 cell proliferation, with the exception of the highest dose, namely 40 mg/L (Figure 4b). Cell viability was similarly unchanged at all concentrations analyzed (Figure 4d). The quantification of Caspases 3/7, after cellular treatment with gallic and ferulic acids, respectively, showed no significant difference compared to the untreated control (Figure 4e. The potential effects of gallic and ferulic acids (20 mg/L, respectively) on the migration of human monocytes were studied using Boyden chambers, in either the presence or absence of Caco2 cells (Figure 4f,g). The number of migrated cells were statistically higher in the presence of Caco2 with gallic and ferulic acids, compared to both the untreated control and treated samples without intestinal cells.

### 3.5. Linear Discriminant Analysis (LDA)

The entire data set was analyzed by LDA, in order to explain the observed response variability of the investigated cell lines (Caco2, L929, U937) as a function of the treatment with different gallic and ferulic acid doses (0, 2.5, 5, 10, 20 and 40 mg/L). Figure 5a shows a scatter plot of the 10 investigated treatments on the space defined by the first two canonical functions (Root 1, Root 2), accounting for 70% and 23% of the total variability, respectively. The multivariate technique showed high discrimination power as indicated by the Wilks lambda value (0.00114), significant at *p <* 0.00001. The scores of the cases showed that the different treatments were grouped separately. As revealed from the values of canonical functions standardized within variance, the distribution of the cases along the positive branch of Root 1 was strongly influenced by the vitality tests on L929 and U937, the MTT test on L929, and the migration test (values of the first canonical discriminant function equal to 1.35, 0.81, 1.01, 1.21, respectively) (Figure 5b). The gallic and ferulic acid treatments, between 2.5 and 10 mg/L, were scored in the right quadrant of the scatterplot (positive Root 1, positive Root 2), opposite to the 20 and 40 mg/L treatments on the left quadrant (negative Root1, negative Root 2) of the scatterplot. The untreated control was positioned in the midstream, between the low dose (2.5–10 mg/L) and high dose (20–40 mg/L) groups, respectively. The high dose group is strictly associated with the negative branch of the first component (influenced by the MTT test on U937, with a value of the first canonical discriminant function equal to −1.01). The score of the cases along the second canonical function (Root 2) was mainly influenced by the vitality of Caco2 (value of the second canonical discriminant function equal to 1.22). In addition, LDA showed that all the high dose treatments were completely discriminated (discrimination percentage of 100% at *p* > 0.05), while the low dose treatments were only partially discriminated (mean discrimination percentage of 88.3% at *p* = 0.09).

### 3.6. Effect of Gallic and Ferulic Acids on Intestinal Tridimensional Models

The effects of gallic and ferulic acids on the viability and toxicity of the three principal cellular components of the intestinal wall have been studied by using the cell lines individually. To better recapitulate what could happen in the gut wall tissue, an intestinal tridimensional model has been developed. In this model, all three cell populations previously studied were grown together, supported by a collagen type I matrix, able to recreate a support which mimicked the dermis. The tridimensional models were treated for 24 h with both gallic and ferulic acids, at two doses of 5 and 20 mg/L, respectively. Thereafter, tridimensional models were paraffin-embedded and stained (Figure 6). Histological analysis of the mucosal layer showed macroscopic impairment of the mucosal structure, and a down-regulation of occludin expression, with 20 mg/L gallic acid (Figure 6b) compared to the control. In contrast, when using 5 mg/L gallic acid (Figure 6b), there was an up-regulation of the expression of the tight junction occludin (Figure 6c) in treated mucosa, compared to the control. Samples treated with 20mg/L gallic acid significantly down-regulated both the thickness layer and the expression of occludin. No significant effect was elicited by ferulic acid on the epithelial layer structure at either 5 or 20 mg/L (Figure 6b). Again, a decreased expression of occludin was detected in the samples treated with 20 mg/L ferulic acid, compared to the untreated control. No effect on occludin expression was detected with 5 mg/L ferulic acid. A similar effect on occludin expression was observed when intestinal equivalents were performed using the human health cell line NCM460, instead of Caco2 (Appendix A). Thiazine staining showed that 20 mg/L gallic and ferulic acids, respectively, induced migration of monocytes in the direction of the epithelial layer, where the integrity was affected.

## 4. Discussion

The aim of the present study was to evaluate the effect of two distinct phenolic compounds, using a new in vitro intestinal wall model in order to understand whether polyphenol effects on humans were related to their concentration. While it is well-known that in vitro models are limited for the testing of phenolic compounds, as the latter are mainly metabolized by intestinal microflora, a direct effect of these compounds on intestinal mucosa cannot be excluded. Intestinal microbiota can vary considerably from individual to individual, and may be severely compromised in people suffering from chronic intestinal diseases such as Inflammatory Bowel Disease (IBD) [18]. We tested the effect of gallic acid and ferulic acid on three different cell lines (colonocytes, fibroblasts and monocytes) and on a tridimensional model of intestinal wall tissue, composed of a mucosal layer and a submucosa with fibroblasts and monocytes, respectively. We selected the most represented compound from fruits (gallic acid) and cereals (ferulic acid) for our experiments.

### 4.1. Gallic and Ferulic Acids and Intestinal Tissue Cell Lines

The present study clearly showed that the differing results obtained were dependent on the cell line used, as well as the polyphenol concentrations. In fibroblasts, no effect on cell proliferation was elicited by either gallic or ferulic acids at any of the concentrations tested. Moreover, no difference in caspase 3/7 activation was found between treated cells and control cells. Ferulic acid was shown to significantly improve cell viability, whereas no effect was obtained with gallic acid. Interestingly, both polyphenols were toxic when administered at a dose of 40 mg/L. Using the wound-healing assay, it was possible to analyze the capacity of gallic and ferulic acids to induce fibroblasts to heal an in vitro wound. This process is attributable to a precise balance between the migration, proliferation and differentiation of cells near the damaged area [19]. The fibroblasts represent the cell population, which is directly involved in the production of the dermis matrix and has the ability to heal skin wounds [20]. On fibroblasts, ferulic acid was able to promote wound-healing at 5, 10 and 20 mg/L, but inhibited cell migration at the dose of 40 mg/L. Gallic acid stimulation improved wound-healing at 2, 5 and 5 mg/L, but similarly induced toxicity at the 40 mg/L dosage. Caco2 cell lines originate from adenocarcinoma, and are the subject of numerous publications promoting their use as biological reference models for the study of enterocytes in vitro [21]. In the present experiments, gallic acid induced toxicity by inhibiting proliferation at 40 mg/L, and had an effect on cell viability starting from 20 mg/L. Instead, ferulic acid was shown to be toxic at a lower dose of 10 mg/L, by inhibiting proliferation. From 10 to 20 mg/L, ferulic acid was toxic as it inhibited cell viability, whilst surprisingly, no effect was detected using the 40 mg/L dose. Moreover, these data were confirmed by caspase 3/7, which showed a similar trend, suggesting that, in Caco2 cells, the down-regulation of proliferation and viability could be mediated by the induction of cell death by activating the apoptotic process. In the intestinal wall tissues are components of innate immunity, such as monocytes and dendritic cells, which are able to respond to oxidative stress and infections [22]. Using the U937 monocyte cell line, gallic acid was shown to up-regulate monocyte activation, starting from the 5 mg/L dose, with no effect on cell viability. Ferulic acid was shown to have no effect on proliferation except at 40 mg/L, and no effect on cell viability was detected. No difference in caspase 3/7 activation was found between treated cells and control cells. Unlike the other cell lines analyzed, the viability of U937 cells were largely unaffected when treated with the two phenolic acids. This suggests that, as a consequence of the immunity nature of these cells, an inflammatory or stressful stimulus may act to induce a greater proliferation of these cells. To investigate a possible interaction between epithelial cells and monocytes, a migration assay was performed using U937 cells alone, and U937 cells co-cultured with Caco2 cells. Interestingly, cell migration was observed only when U937 and Caco2 were co-cultured. This data suggests that gallic and ferulic acid may induce monocyte migration through the signals released by Caco2 cells, which showed apoptotic activation under stimulation of both polyphenols at 20 mg/L [23]. The LDA analysis showed that, taken together, from all the results obtained from the analysis of the cell lines cultured in the monolayer, it was possible to clearly discriminate all the high dose treatments (20 and 40 mg/L) of both gallic and ferulic acids, which produced negative effects on cells, from the low dose treatments (2.5, 5 and 10 mg/L). The low dose treatments were only partially discriminated, but with overall positive effects on cell health.

### 4.2. Gallic and Ferulic Acids and Tridimensional Intestinal Equivalent Model

Once the cells were studied individually, the effects of gallic and ferulic acids were tested on a three-dimensional system composed of all three cell lines that were previously analyzed singularly. The three-dimensional cell system, in which several cell lines were assembled to constitute an artificial tissue, represents an alternative method to both in vivo mouse models and in vitro bidimensional cell cultures [24]. Interestingly, a different effect was observed when polyphenols were individually tested in this model. First of all, when analyzed as part of a complex tissue, treatment with 20 mg/L gallic acid induced macroscopic histological damage with lesions similar to ulcerations. These lesions may be related to a significant decrease in the occludin expression observed. Ferulic acid only induced a significant decrease in occludin expression, but both stimuli induced monocyte migration close to the epithelial layer. These effects were not detectable at doses of 5 mg/L. At this concentration, gallic acid seemed to improve intestinal permeability, by increasing epithelial thickness and occludin expression compared to the control. Basically, the results obtained with the three-dimensional model allowed us to set the toxicity of these compounds at a lower dose than that suggested by tests conducted on monolayer models, which also showed varying responses at different treatments depending on the cell line.

### 4.3. Conclusions

All these data, taken together, suggested that gallic and ferulic acids can exert toxic effects on the intestinal wall, if present at high concentrations. The concentrations of both gallic and ferulic acids normally consumed in the diet can vary. Sosulski and coworkers [25] found that in flour, directly after milling, the trans-ferulic acid content was 63.6 μg/g, while in flour stored for 6 months, the level was 23.3 μg/g. Keskin-Šašić et al. showed that gallic acid concentration can vary from 7.3 mg GAE/100 mL in aloe vera juice to 71.8 mg GAE/100 mL in cranberry juice [26]. From the present investigation, we found that the two commercially available supplements analyzed displayed concentrations of gallic and ferulic acids that were significantly higher than the amounts present in the respective flour or fruit sources. In the present study, no in vivo assays have been performed, thus it is not possible to predict the real role of the gut microbiome during the absorption mechanism in the intestinal tract for the phenolic compounds analyzed. Moreover, the in vitro intestinal models used in the present work only simulate what happens in vivo. Taken together, our results suggest the idea that high ingestion of polyphenols could induce negative effects on the intestinal wall’s integrity. However, more studies, especially with in vivo models and pre-clinical or clinical studies that include the metabolism due to the gut microbiome, are needed to more firmly confirm this hypothesis.

## Figures and Tables

**Figure 1 nutrients-12-01591-f001:**
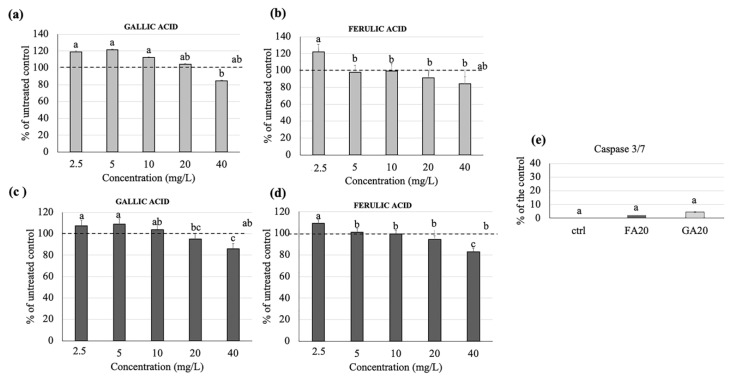
Gallic and ferulic acid effects on L929 mouse fibroblast proliferation and viability. L929 fibroblasts were treated with gallic and ferulic acids at 2.5, 5, 10, 20 and 40 mg/L, respectively. The MTT (3-(4,5-Dimethylthiazol-2-Yl)-2,5-Diphenyltetrazolium Bromide) assay (**a,b**) and viability assays (**c,d**) were performed 24 h after treatment. For caspase 3/7 analysis, L929 cells were treated with gallic and ferulic acid at the concentration of 20 µg/mL or diluent as the control. After 24 h, cells were stained with CellEvent™ Caspase-3/7 Green Detection Reagent (**e**). Data are expressed as a percentage of the control. Statistical analysis was performed and, between brackets, different letters indicate mean values that are significantly different at *p* < 0.05.

**Figure 2 nutrients-12-01591-f002:**
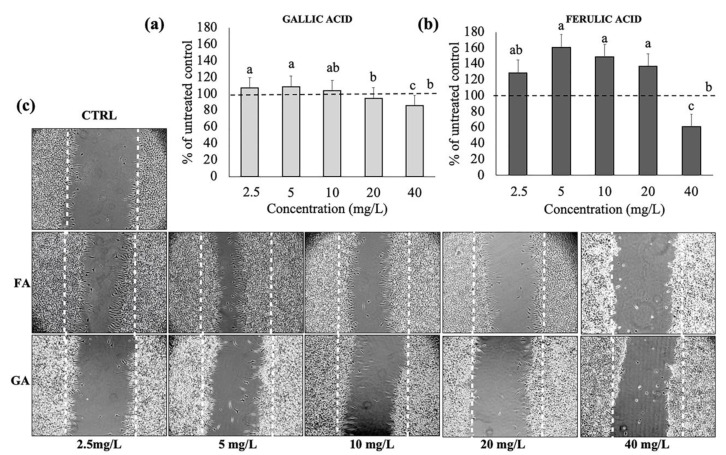
Wound-healing assay using L929 with gallic and ferulic acid. (**a**) number of fibroblasts migrated after treatment with gallic and (**b**) ferulic acids at 2.5, 5, 10, 20 and 40 mg/L. (**c**) Significant pictures of the migrated cells. Data were expressed as percentage compared to the control. Statistical analysis was performed, and between brackets, different letters indicate mean values that are significantly different at *p* < 0.05.

**Figure 3 nutrients-12-01591-f003:**
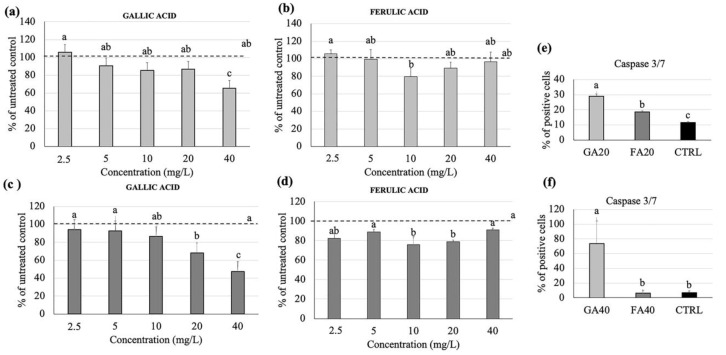
Gallic and ferulic acid effects on Caco2 intestinal cell proliferation and viability. Caco2 cells were treated with gallic and ferulic acids (2.5, 5, 10, 20 and 40 mg/L), respectively. The MTT (**a,b**) and viability assays (**c,d**) were performed 24 h after treatment. For caspase 3/7 analysis, Caco2 cells were treated with gallic and ferulic acids at either 20 mg/L (**e**) or 40 mg/L (**f**) and diluent as the control. After 24 h, cells were stained with CellEvent™ Caspase-3/7 Green Detection Reagent. Data are expressed as a percentage of the control. Statistical analysis was performed, and between brackets, different letters indicate mean values that are significantly different at *p* < 0.05.

**Figure 4 nutrients-12-01591-f004:**
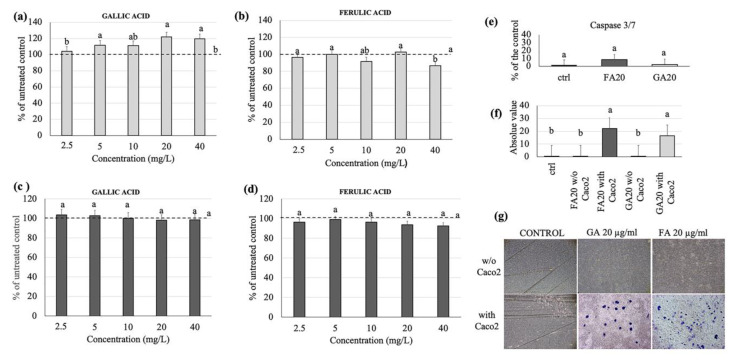
Gallic and ferulic acid effects on U937 monocytes cell proliferation and viability. U937 cells were treated with gallic and ferulic acids (2.5, 5, 10, 20 and 40 mg/L) respectively. The MTT (**a,b**) and viability assays (**c,d**) were performed 24 h after treatment. For caspase 3/7 analysis, U937 cells were treated with gallic and ferulic acids, respectively, at 20 mg/L or diluent as the control. After 24 h, cells were stained with CellEvent™ Caspase-3/7 Green Detection Reagent (**e**). U937 cell migration was measured as indicated in the Materials and Methods section, either with or without Caco2 cells. The mean number ± SD of migrated cells were counted and data are expressed as percentage of the control (**f,g**). Statistical analysis was performed, and between brackets, different letters indicate mean values that are significantly different at *p* < 0.05.

**Figure 5 nutrients-12-01591-f005:**
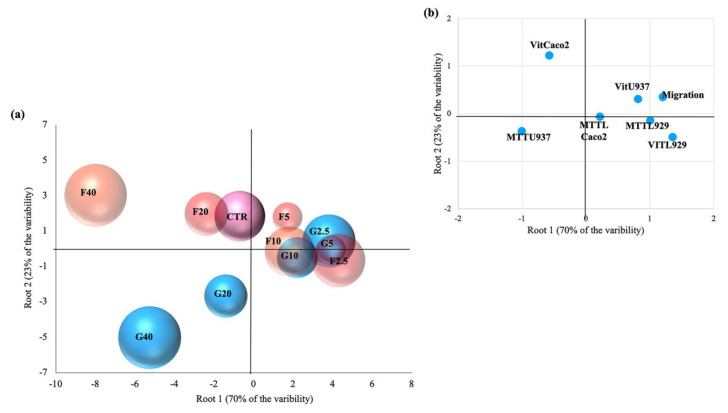
Scatter plot of the 10 investigated treatments, namely (**a**) gallic and ferulic acids, at doses of 2.5, 5, 10, 20 and 40 mg/L, respectively, which were defined by the first two canonical functions of linear discriminant analysis and (**b**) projection of variables (as reported in the material and methods section) on the first two dimensions.

**Figure 6 nutrients-12-01591-f006:**
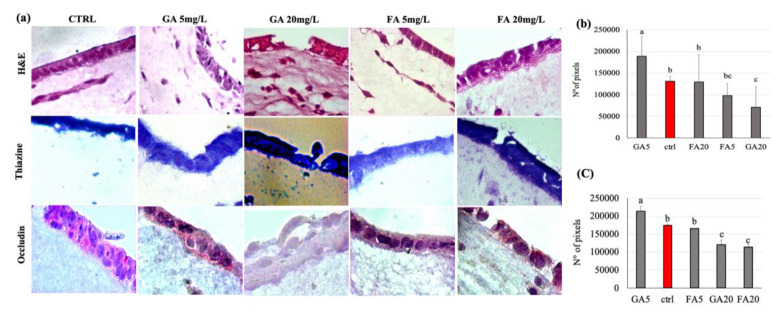
Intestinal equivalents obtained by seeding Caco2 cells on dermal equivalents that were paraffin-embedded. (**a**) Sections were stained with hematoxylin and eosin (H&E), thiazine and occludin markers; Fast Red was used as a chromogen. (**b**) The thickness of the epithelial layer was measured by image pixel count using ImageJ 2.0.0-rc-69/1.521 (**c**) Stained areas were evaluated by image pixel count using ImageJ 2.0.0-rc-69/1.521. Experiments were conducted in triplicate from different samples. Statistical analysis was performed, and between brackets, different letters indicate mean values that are significantly different at *p* < 0.05.

**Table 1 nutrients-12-01591-t001:** Gallic and ferulic acid concentration analysis in two commercially available dietary supplements: Blueberry supplement and Supplement A.

	Gallic Acid (mg/g) ± s.d.	Ferulic Acid (mg/g) ± s.d.
**Blueberry Supplement**	0.3252 ± 0.030	1.3695 ± 0.140
**Supplement A**	0.6354 ± 0.035	1.8494 ± 0.180

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
