# Peer review of "Are Supplements Safe? Effects of Gallic and Ferulic Acids on In Vitro Cell Models"

_nutrients, 2020, doi:10.3390/nu12061591_

Round 1

Reviewer 1 Report

The paper is interesting, however, there are some points that need to be clarified or supplemented before publication.

  • Manuscript should be prepared according to “Guide for Authors” i.e. Keywords the font should be changed; line 97 – “0.86 mL min -1” should be: 0.86 mL/min;
  • Please provide the origin of the samples (supplements);
  • What was the purity of the standards (gallic acid and ferulic acid)?
  • There is no information about validation of HPLC-MS/MS method;
  • Exemplary chromatogram of analyzed compounds should be provided;
  • The Authors should familiarize themselves with the proper format for References, make appropriate corrections. For example, “Tsao, R. Chemistry and biochemistry of dietary polyphenols. 2010 Dec 2; 1231-46. doi: 42110.3390/nu2121231. Epub 2010 Dec 10. Review” should be: Tsao, R. Chemistry and biochemistry of dietary polyphenols. Nutrients 2010, 2, 1231-1246, doi: 42110.3390/nu2121231.

Author Response

Response to Reviewer 1 Comments

Point 1: Manuscript should be prepared according to “Guide for Authors” i.e. Keywords – the font should be changed; line 97 – “0.86 mL min -1” should be: 0.86 mL/min;

Response: line 28: the font of the keywords has been changed, according the “Guide for Authors”, line 100:  0.86 mL min -1” has been changed in 0.86 mL/min;

Point 2: Please provide the origin of the samples (supplements);

Response: line 82: the origin of the supplements has been added. Only for the reviewer: the food supplements we analyzed are marketed and sold in Italy in pharmacies, based on the presence of a single raw material in addition to some excipients. The raw material of the first one is a dry extract obtained from a particular apple cultivar (called Annurca), with a high polyphenol content, sold for the control of hypercholesterolemia. The second one is based on dry blueberry extract, sold for its antioxidant activity and for possible beneficial effects on vision and retina. Although it is rather easy to find out which supplements are by using the Internet, we do not think it is appropriate to indicate them with their trade name as it could be a positive or negative advertisement for the companies that actually market them. Our focus was not to test the effects or the efficacy of food supplements sold on the market, but simply to reason that food supplements that commonly could have a high title in the two active molecules that our study is interested in, from the point of view of their in vitro effects on a model of intestinal barrier;

Point 3: What was the purity of the standards (gallic acid and ferulic acid)?

Response: line 76, 77: the purity of standards has been added;

Point 4: There is no information about validation of HPLC-MS/MS method.

Response: line 95 and lines 495 - 501: authors added references about validation of HPLC-MS/MS method;

Point 5: Exemplary chromatogram of analyzed compounds should be provided;

Response: in the supplementary results file, authors added supplementary result 1, which represents the chromatograms of the samples analyzed;

Point 6: The Authors should familiarize themselves with the proper format for References, make appropriate corrections. For example, “Tsao, R. Chemistry and biochemistry of dietary polyphenols. 2010 Dec 2; 1231-46. doi: 42110.3390/nu2121231. Epub 2010 Dec 10. Review” should be: Tsao, R. Chemistry and biochemistry of dietary polyphenols. Nutrients 2010, 2, 1231-1246, doi: 42110.3390/nu2121231.

Response: from line 443 to 527: references have been corrected according the proper format of the journal.

Reviewer 2 Report

In the current report authors investigate the effects of two polyphenols, gallic and ferulic acid present in two different supplements, in terms of safety, as they are widely used for their health-promoting properties. Specifically they have tried to understand whether polyphenol effects on humans was related to their concentration, as stated in the beginning of the discussion. In order to do that, authors have used a new in nitro model of intestinal wall and three different cell lines, with the specific aim to evaluate any undesirable effects of both polyphenols used in high concentrations. Although methodology is correct, my main concern-doubt is how data provided by the authors are applicable (in terms of safety investigation) since they have used normal mouse fibroblasts (L929) and two different cell lines CaCo2 (colorectal adenocarcinoma) and U937 (leukemic cells), both malignant. Results could have scientific interest, if all cell lines were normal and not malignant, simulating, at least in part, the effects of both polyphenols on intestinal wall.

Author Response

Point 1: In the current report authors investigate the effects of two polyphenols, gallic and ferulic acid present in two different supplements, in terms of safety, as they are widely used for their health-promoting properties. Specifically they have tried to understand whether polyphenol effects on humans was related to their concentration, as stated in the beginning of the discussion. In order to do that, authors have used a new in nitro model of intestinal wall and three different cell lines, with the specific aim to evaluate any undesirable effects of both polyphenols used in high concentrations. Although methodology is correct, my main concern-doubt is how data provided by the authors are applicable (in terms of safety investigation) since they have used normal mouse fibroblasts (L929) and two different cell lines CaCo2 (colorectal adenocarcinoma) and U937 (leukemic cells), both malignant. Results could have scientific interest, if all cell lines were normal and not malignant, simulating, at least in part, the effects of both polyphenols on intestinal wall.

Response: Authors agree with the reviewer, given that if all cell lines used were normal, and not malignant, the models maybe recapitulated the intestinal wall better than what we obtained. However, authors decided to use Caco2 cells for the intestinal wall, because it has been showed that they differentiate spontaneously in culture without supplementation of differentiating factors and they have been extensively used as a model of the intestinal barrier for in vitro toxicology studies (Artursson et al., 2001; Sambuy et al., 2005). Caco2 cells develop morphologic characteristics of normal enterocytes when grown on plastic dishes or nitrocellulose filters and they can be used for permeability assay cells as they grow on collagen-coated polycarbonate membranes, that represent a valuable transport model system (Hidalgo et al 2011). Authors performed preliminary data by using human healthy colonocyte cell line (NCM460) in order to confirm that the in vitro model used was correct: in the supplementary figure 2, authors added the intestinal equivalents performed with NCM460 instead of the Caco2. Hematoxylin and eosin showed that gallic acid treatment reduced the integrity of the intestinal mucosa at the dose of 20 mg/L as well as what happens when Caco2 were used for the intestinal model. This effect is less evident when gallic acid was added at the concentration of 5mg/L. Tight junction occludin is also less expressed in the samples treated with gallic acid at the concentration of 20 mg/L, compared to untreated control. Authors added a comment in the main test, line 340. Given that Caco2 appeared to better replicate the intestinal wall in vitro than NCM460, authors decided to use only Caco2 cells. Regarding U937 cells: even if they are leukemic cells, they are monocytes which are still able to differentiate in macrophages in vitro. For this reason, this specific cell line is usually used, especially in co-cultivation with neighboring cells, as it can be an option to mimic the relevant interactions between cells and their surroundings, as in natural tissues happen (Chanput 2015).  

Reviewer 3 Report

The manuscript titled “Are supplements safe? Effects of gallic and ferulic acids on in vitro cell models” evaluates the unsafety caused by the phenolic compounds, such as gallic acid and ferulic acid, are remained without metabolized in the body. Furthermore, this study for solving clinical difficulties was proposed through the application of in vitro technique, which is considered to be a very significant study as a result. From my point of view, there are major concerns that avoid this manuscript to be accepted for publication and there are additional experiments that are required to sustain the conclusions achieved by the authors.
 Major Concerns:
1.     In most cases, GA or FA is mixed with various additives, etc. and consumed as a mixture. Therefore, rarely ingest GA or FA in pure content. In the end, it may be considered that there is sufficient potential for side effect control by the added substances and there is also a complementary effect by other compounds. To mention side effects by over-dose consumption or unmetabolizing FA and GA, it is thought that the in vitro test results are very fragmentary, so it is necessary to compare the existing conventional product with the FA or GA that contains the same or higher content as the experiment applied.

2.     In addition, it is necessary to compare the two products mentioned in Table 1 with the approximate content ratio of FA and GA as well as other components. Because there is a possibility of easily excluding the efficacy of other components, there is a lack of basis for deleting other components. Therefore, it is necessary to reconcile the results and discussions based on the overall content ratio, as it could be considered that there is a enough chance of causing an analytical error if it is simply limited to the comparison of two components.

3.     According to the experimental concentration, 40 mg/L was treated in each cell slightly different, but MTT and cell viability were likely to be very seriously concerned. However, existing studies have shown that 40 mg/L is already predictable enough to induce a cell survival changes because 40 mg/L was mentioned as a very high concentration in vitro experiments. In addition, for FA and GA, treatment above the 10 ug/ml level has already been reported to cause differences in statistical significance in cell viability, although the cell types are different. After all, the author's opinion should be included in the discussion what it would mean to have no clinical claim, even though in vitro results are already fully predictable.

Minor Concerns:
1.      There are lacks of rationale for why the each experiment was applied before the briefing on the results of each experiment.
 2.     According to wound healing assay results in Figure 2, the physiological results on figure 2(C) and the percentage of transferred cells depending on the concentrations of (a) GA and (b) FA doesn’t reconcile. How the authors explain this fact?
 3.     Please check if Figure 5(b) “VITL929” is correct.
 4.     In order to secure a clear key point that the author intends to propose, paragraph classification of the Discussion part is required in content.

Author Response

Response to Reviewer 3 Comments

Major Concerns:

Point 1.     In most cases, GA or FA is mixed with various additives, etc. and consumed as a mixture. Therefore, rarely ingest GA or FA in pure content. In the end, it may be considered that there is sufficient potential for side effect control by the added substances and there is also a complementary effect by other compounds. To mention side effects by over-dose consumption or unmetabolizing FA and GA, it is thought that the in vitro test results are very fragmentary, so it is necessary to compare the existing conventional product with the FA or GA that contains the same or higher content as the experiment applied.

Response: We completely agree with Reviewer 3, since polyphenol-based food supplements are always a very complex mixture of active molecules, including FA and GA, with possible complementary or synergistic effects. Our study, however, does not aim to analyze the possible toxic effects of some particular polyphenol-based food supplements, but to highlight that there are potential phenomena of alteration of the intestinal barrier induced by FA and GA, substances certainly present in food supplements, with not negligible dosages. For this reason, we have modified the conclusion sentence in line 432 - 438. Despite this, when one wants to analyze the effects of single molecules at the organ level, it is a common experimental procedure to analyze their toxic or physiological effects on cellular models. This is exactly what we have done, with all the limitations of an in vitro model, but with the advantages of a 3D multicellular model that better represents what happens in the intestine respect to monolayers of any cell line.

Point 2.    In addition, it is necessary to compare the two products mentioned in Table 1 with the approximate content ratio of FA and GA as well as other components. Because there is a possibility of easily excluding the efficacy of other components, there is a lack of basis for deleting other components. Therefore, it is necessary to reconcile the results and discussions based on the overall content ratio, as it could be considered that there is a enough chance of causing an analytical error if it is simply limited to the comparison of two components. 

Response: The two raw materials we have analyzed are both commercial dry extracts of fruits or berries. They are mixtures of thousands of different molecules whose reciprocal relationships change during production from batch to batch. Whatever in vitro effects we observed, we could not have attributed it to any particular chemical compound present into the extract. If the supplements had the same effect as FA and GA, or if we had observed very different effects, we could not have drawn any further conclusions, not being able to exclude synergies or complementarities between the hundreds of active compounds present into these extracts. The goal of our research work was not to test the effects or efficacy of one or more complex food supplements. This should be done on animal models or directly on humans, since they are products on the market, already authorized for use on humans. The novelty of this work is that of having verified non-positive effects of high concentrations of FA and GA in vitro, and to underline the possible risks of high dosages of supplements that contain high levels of GA or FA, highlighting the possible cellular mechanisms at the basis of an alteration of the intestinal permeability. The verification of these possible risks needs to be done by using in vivo models, in which the intestinal microbiota is present, which represents the true variable that cannot be analyzed by any in vitro study.

Point 3.     According to the experimental concentration, 40 mg/L was treated in each cell slightly different, but MTT and cell viability were likely to be very seriously concerned. However, existing studies have shown that 40 mg/L is already predictable enough to induce a cell survival changes because 40 mg/L was mentioned as a very high concentration in vitro experiments. In addition, for FA and GA, treatment above the 10 ug/ml level has already been reported to cause differences in statistical significance in cell viability, although the cell types are different. After all, the author's opinion should be included in the discussion what it would mean to have no clinical claim, even though in vitro results are already fully predictable.

Response: We agree that the concentration of 40mg/L is to be considered high, but not unattainable inside the intestine or stomach, after taking a "normal" dose of food supplements, taken on an empty stomach and with a sip of water, as often happens. Although other studies have indicated alterations of cell viability at the dose of 10mg/L, the novelty of our study is to have shown that the effects change from single cell to a more complex 3D model, and that overall a mechanism of possible alteration of intestinal permeability, induced by GA and FA exists. For what could happen in vivo and the clinical significance, however, we have no plausible hypotheses, since, as already mentioned, we need in vivo pre-clinical or clinical studies that include the metabolism due to the gut microbiome.

Minor Concerns:

Point 1.    There are lacks of rationale for why the each experiment was applied before the briefing on the results of each experiment.

Response:  The authors added the rationale of each experiment before the briefing of the results. In particular, authors added a comment in line 205-207, 219-221, 253-255, 271-272, 325-330.  

Point  2.    According to wound healing assay results in Figure 2, the physiological results on figure 2(C) and the percentage of transferred cells depending on the concentrations of (a) GA and (b) FA doesn’t reconcile. How the authors explain this fact?

Response: We agreed with the reviewer that sometimes the results on figure2 (c) appear not to reconcile with the percentage in figure (a) and (b). The pictures are representative of the counting of the cells, which was obtained from the analysis of at least 3 pictures of each condition, but probably it is difficult to detect or not migrating cells because of the low quality of the pictures presented. For this reason, the quality of figure 2 (c) has been improved in order to make more evident the presence or not of the migrating cells.

Point 3.     Please check if Figure 5(b) “VITL929” is correct

Response: Figure 5(b) “VITL929” is correct has been corrected.

Point 4.     In order to secure a clear key point that the author intends to propose, paragraph classification of the Discussion part is required in content.

Response: In order to secure a clear key point that the author intended to propose, the discussion has been divided in different paragraphs, as suggested by the reviewer.

Round 2

Reviewer 1 Report

Authors improved their manuscript according to reviewer's comments. I can confirm that all of my comments have been taken into an account and all appropriate places were successfully corrected of explained. So my recommendation could be accept.

Reviewer 2 Report

Following several improvements throughout the manuscript and clarification regarding the choice of different cell lines used in several experimental procedures, my only recommendation is to include in the manuscript a small part stating the rationale (as in authors' response including referenses as provided) of using CaCo2 and U937 cells, in order to be this choise clear also for the readers.

Minor comments:

1) Line 282: Please correct Figure 4d to Figure 4e      (Caspases 3/7 Assay)

2) Line 284: Please correct Figure 4d-e to Figure 4f-g (Migration Assay) 

3) It would be helpful include in each figure legend the meaning of the              different letters used in statistical analysis.

Reviewer 3 Report

No more considerations.